# Pelvic Pain, Mental Health and Quality of Life in Adolescents with Endometriosis after Surgery and Dienogest Treatment

**DOI:** 10.3390/jcm12062400

**Published:** 2023-03-20

**Authors:** Elena P. Khashchenko, Elena V. Uvarova, Vladimir D. Chuprynin, Margarita Yu. Pustynnikova, Timur Kh. Fatkhudinov, Andrey V. Elchaninov, Zhanna R. Gardanova, Tatyana Yu. Ivanets, Mikhail Yu. Vysokikh, Gennady T. Sukhikh

**Affiliations:** 1FSBI “National Medical Research Center for Obstetrics, Gynecology and Perinatology Named after Academician V.I. Kulakov” Ministry of Healthcare of the Russian Federation, 4, Oparina Street, 117997 Moscow, Russia; 2Department for Obstetrics, Gynecology, Perinatology and Reproduction, Sechenov First Moscow State Medical University, Trubetskaya Str. 8, Bld. 2, 119991 Moscow, Russia; 3Faculty of Fundamental Medicine, Moscow State University Named after M.V. Lomonosov, 119991 Moscow, Russia; 4Department of Histology, Cytology and Embryology, Peoples’ Friendship University of Russia (RUDN), Miklukho-Maklaya Str. 6, 117997 Moscow, Russia; 5A.N. Belozersky Research Institute of Physico-Chemical Biology MSU, Leninskye Gory, House 1, Building 40, 119992 Moscow, Russia

**Keywords:** endometriosis, adenomyosis, dysmenorrhea, pelvic pain, psycho-emotional status, quality of life, depression, anxiety, progestins, dienogest, adolescents, treatment

## Abstract

Background: Diagnostic and treatment delays have caused significant impacts on the physical and emotional well-being of adolescents with endometriosis, though such research is limited. This study aimed to assess the effects of one-year dienogest therapy on the clinical picture, pain patterns, psycho-emotional status, and quality-of-life indicators in adolescents with endometriosis after surgical treatment. Methods: The study enrolled 32 girls aged 13–17 with peritoneal endometriosis to analyze one-year dynamics of the Visual Analog Scale (VAS), McGill Pain Questionnaire, Beck Depression Scale (BDI), Hospital Anxiety and Depression Scale (HADS), Spielberger State-Trait Anxiety Inventory (STAI) and SF-36 quality-of-life survey scores along with clinical and laboratory indicators before surgery and after one-year dienogest therapy. Results. The therapy provided a significant alleviation of endometriosis-associated clinical symptoms, including dysmenorrhea, pelvic pain, gastrointestinal/dysuria symptoms, decreased everyday activity (<0.001), a decrease in anxiety/depression scores (BDI, HADS, STAI), and quality-of-life improvement (<0.001). These effects were accompanied by beneficial dynamics in hormone and inflammatory markers (prolactin, cortisol, testosterone, estradiol, CA-125, neutrophil/lymphocyte ratio; <0.005) within reference ranges. A low body mass index and high C-reactive protein levels were associated with higher VAS scores; a high estradiol level was a factor for anxiety/depression aggravation (<0.05). Conclusions: Dienogest, after surgical treatment, significantly improved quality of life and reduced pain symptoms while showing good tolerability and compliance, and reasoning with timely hormonal therapy in adolescents with endometriosis.

## 1. Introduction

Clinical symptoms of endometriosis (EMS) develop starting from puberty, though in rare cases, the disease can manifest long before menarche [1]. Epidemiological data on EMS in adolescents are sparse [2]; EMS is considered one of the leading causes of secondary dysmenorrhea and chronic pelvic pain in adolescents [3,4]. EMS has been diagnosed in 60% of teenage girls undergoing diagnostic laparoscopy for persistent pelvic pain and in 75% of those with drug-resistant dysmenorrhea [5]. The EMS-associated pain in adolescents includes menstrual pain, non-cyclic pelvic pain, and defecation/urination pain, as well as dyspareunia in sexually active girls [6]. Despite the obvious psychological and social burden of these symptoms, dedicated studies on the psycho-emotional status of adolescents treated for genital EMS are limited [4,7].

Chronic pain substantively contributes to affective disorders; in particular, chronic pelvic pain has been pathogenetically linked to depression [8,9]. Persistent pain sensations characteristic of EMS promote secondary psychoneurological dysfunctions (anxiety, depression), which further aggravate the individual perception of noxious stimuli [10,11,12]. The resulting higher rates of neuroticism, hypochondria, emotional lability, and decreased quality of life in patients with EMS have been confirmed by a large amount of research [8,9,10,11,12,13].

Adolescents diagnosed with a chronic and painful disease, in particular, experience increased emotional vulnerability and difficulties in coping with the illness, causing frustration, depression, and a lack of understanding of their condition along with physical, hormonal, and emotional changes related to the crucial time of puberty [14]. Self-esteem, social involvement, school attendance, and performance are critically affected in young patients with EMS-associated symptoms, especially because of the prolonged time of diagnostic delay [15,16]. In this regard, a particularly relevant issue is the early start of pathogenic therapy from the age of disease manifestation. Though there is a discussion regarding whether early diagnosis and treatment in teenage endometriosis prevent disease progression, it is accepted that the effective treatment of endometriosis in adolescents is necessary to reduce pain symptoms as well as to improve quality of life [4]. However, there is a lack of research on the risk-benefit profile of EMS treatment in adolescents and quality of life parameters in the literature.

A relationship between mental health and inflammatory status was evidenced in a number of studies [17,18,19,20,21]. The association of ‘affective’ symptoms with high levels of inflammatory markers was considered reciprocal: inflammation can promote mood disorders and vice versa, potentiating positive feedback [22].

Systemic inflammation is a core link in EMS pathogenesis. Estradiol is known to directly stimulate prostaglandin E2 synthesis, thereby enhancing noxious stimuli in a COX2-dependent manner [23]. Many studies associate EMS with increased levels of IL-1β, IL-6, IL-8, TNF-α, CCL-2, CCL-5, and VEGF in systemic circulation and peritoneal fluid [3,24,25,26]. Due to the central role of pro-inflammatory cytokines in the clinical picture of the disease, non-steroidal anti-inflammatory drugs (NSAIDs) remain first-line painkillers in their management.

Amid the lack of unified opinions on whether inflammation should be considered a primary cause of EMS or a secondary factor in its transition to the chronic phase [27], there are distinct pathogenetic parallels between EMS and pain syndrome-associated affective disorders. EMS is treated similarly in adolescent and adult patients, with an emphasis on the long-term prognosis and preservation of reproductive potential in younger individuals [6]. Importantly, the treatment must ensure positive dynamics for psycho-emotional and quality-of-life indicators [12]. In this study, we aimed to assess the preservation and dynamics of mental health indicators in adolescents treated for peritoneal EMS for one year.

## 2. Materials and Methods

This study aimed to assess the effects of one-year dienogest therapy on the clinical picture, pain patterns, psycho-emotional status, and quality-of-life indicators in adolescents with peritoneal EMS.

The study enrolled 32 patients with a confirmed diagnosis of peritoneal EMS, aged 13–17 at diagnosis, receiving inpatient/follow-up outpatient care at the Pediatric and Adolescent Gynecology Department, Kulakov National Medical Research Center for Obstetrics, Gynecology, and Perinatology. In addition, 43.8% (14/32) of girls had an association with peritoneal endometriosis and diffuse adenomyosis, according to imaging data (US and MRI). The clinical picture and pain patterns were assessed before the commencement and after one year of therapy. Dienogest was used as a first-line conservative pathogenetic option in accordance with the current guidelines (2 mg dienogest daily continuously for 1 year) for postoperative prevention of pain recurrence for at least a year [28]. The menstrual/intermenstrual pelvic pain was relieved symptomatically with NSAID (a non-selective COX-1/2 inhibitor, ≤3 doses a day for ≤5 days).

Inclusion criteria: age 13–17; menstrual history; clinical symptoms of persistent moderate-severe dysmenorrhea or/and chronic pelvic pain resistant to NSAIDs; diagnosis of peritoneal EMS laparoscopically confirmed after the instrumental examination (ultrasound, MRI); laboratory test data availability; written informed consent for the study. The indications for laparoscopy included unsuccessful empirical treatment (NSAIDs, antispasmodics, COCs) or persistent moderate/severe dysmenorrhea with negative imaging results. Data from the laparoscopic picture included surgical diagnosis and the stage of endometriosis according to the revised American Society for Reproductive Medicine (rASRM) criteria. In each case, we preferred an excision of the peritoneal foci; coagulation was performed in less accessible places.

Exclusion criteria: aged over 18; somatic/endocrine/oncological comorbidities; infectious diseases; genital malformations interfering with menstrual outflow; pelvic tumors; mental conditions; laboratory/clinical data missing; informed consent missing.

Study design: retrospective longitudinal observational cohort study. All patients underwent a full clinical/laboratory examination and passed the full survey before and after the one-year dienogest therapy.

Clinical data collected for the study accounted for menstrual history, pelvic pain history (onset, temporal patterns, and specific localization/character of pain sensations), trends in pain intensity since menarche, specific factors aggravating the pains, and the family history of gynecological disorders all contribute.

Laboratory tests included a general blood test: complete blood count (CBC), white blood cells (WBC), red blood cells, platelets, hemoglobin, hematocrit (Hct), and erythrocyte sedimentation rate (ESR). In the biochemical profile, the concentrations of total protein, uric acid, creatinine, direct and total bilirubin, glucose, Ca2+, Fe2+/3+, and highly sensitive C-reactive protein (CRP) in venous blood were determined for all participants. On the 3rd to 4th day of a spontaneous menstrual cycle, all girls included in the study were subjected to an extended analysis of the blood hormonal profile: the levels of the luteinizing hormone (LH), thyroid-stimulating hormone (TSH), thyroxine, follicle-stimulating hormone (FSH), dehydroepiandrosterone sulfate (DHEAS), androstenedione, prolactin (Prl), estradiol (E2), cortisol, testosterone (T), sex hormone-binding globulin (SHBG), and tumor markers Ca-125 and Ca-19-9 were determined. Hormonal assays were carried out by electro- and immunochemiluminiscent methods on Cobas e 411 (F. Hoffmann-La Roche, Basel, Switzer-land), Immulite 2000, and Immulite 1000 (Siemens, Los Angeles, CA, USA) automatic analyzers using reagents of the same companies. The concentrations of anti-Mullerian hormone (AMH) and 17-OH-progesterone (17-OHP) were measured by the enzyme-linked immunoassay on the DYNEX DSX System analyzer and using the Diagnostic Products Corporation (DPC) system on the Immulite device (DYNEX Technologies, Chantilly, VA, USA).

The tests and interview included:(1)Pain scoring (Visual Analog Scale; Short-Form McGill Pain Questionnaire 2);(2)Anxiety/depression surveys (Beck Depression Index (BDI); Spielberger State-Trait Anxiety Inventory (STAI); Hospital Anxiety and Depression Scale (HADS));(3)Quality-of-life scoring (36-Item Short Form Survey);(4)Interview, concerning negative emotional sensations/ psychosomatic complaints/ fatigue (Questions, concerning the heart pain, palpitations, difficulty breathing, fainting, memory problems, paroxysmal headaches, etc.).

The Visual Analog Scale pain scoring (VAS) uses a 10 cm horizontal line segment symbolizing pelvic pain intensity. The patients were asked to place a mark corresponding to their sensations. The distance between the lower end of the segment (‘no pain’) and the mark was measured in cm (points). Pains below 2 points were rated as ‘weak’, 2–4 points as ‘moderate’, 4–6 points as ‘strong’, 6–8 as ‘very strong’, and over 8 points as ‘unbearable’ [14].

A Short-Form McGill Pain Questionnaire 2 (SF-MPQ-2) comprises 22 items exploring 4 domains: constant pain, periodic pain, neuropathic pain, and affective descriptors. The questionnaire includes the list of words to be used for the description of experienced pain qualities and associated symptoms. Each patient was asked to rate the intensity of specific pain sensations experienced over the past week on a scale from 0 to 10, where 0 stood for ‘no pain’, and 10 stood for ‘the worst pain imaginable’ [29].

The Beck Depression Inventory (BDI) is one of the most common tools to assess the severity of depression in adults and adolescents, consisting of 21 items rated on a 4-point rating from 0 to 3. Interpretation is based on the total score and can range from 0 to 63. BDI scores above 9 points indicate depression: mild (sub-depression, 10–15 points), moderate (16–19 points), medium severity (20–29 points), or severe (30–63 points) [30].

The Spielberger State-Trait Anxiety Inventory (STAI) is one of the most widely used tests in clinical settings for the assessment of anxiety as a combination of two components: inherent (a personality trait, STAI-T) and reactive (an immediate psychological state of the subject, STAI-S). It consists of 40 items ranging from Almost Never (0) to Almost Always. The scores are classified as low (≤30 points), moderate (31–44 points), and high (≥45 points); the combined output is used as a measure of the stress burden [31].

The Hospital Anxiety and Depression Scale (HADS) consists of 14 items, each with 4 response options reflecting the degree of symptoms. The anxiety and depression subscales were scored separately. The interpretation involves the total scores for each subscale rated as follows: ≤7, norm (the lack of symptoms); 8–10, subclinical anxiety/depression; ≥11, clinical anxiety/depression [11].

Quality of life (QoL) was assessed using the 36-Item Short Form Survey (SF-36) comprising 36 items grouped into 8 domains: physical functioning, role-related activity, bodily pain, general health, vitality, social functioning, emotional state, and mental health. Each domain earned 0–100 points, and all scores jointly contributed to mental and physical well-being indices [11].

The personal interview included, in addition, questions regarding the characteristics of stressful tension and psychosomatic manifestations. The interview consisted of a list of questions accounting for autonomic nervous system manifestations of subjective stress burden, including cardiovascular symptoms (heart palpitations, sensations of cardiac arrest), respiratory symptoms (air hunger, tachypnea), gastrointestinal symptoms (constipation, diarrhea, bloating, and abdominal pains), fainting, paroxysmal headaches. The test also accounts for neuroticism (degree of anxiety), overall life (dis)satisfaction and difficulties concentrating, sleep characteristics (speed of falling asleep, sensations of strength and depth of sleep, ease of waking up in the morning), as well as in family, school and personal relationships. Patients were asked to answer each question in three ways (often, sometimes, and never) to describe their sensations. Additionally, patients answered the questions about emotional reactivity (nervous flashes), attitudes towards social environments (trust in people, sensation of esteem of subject’s achievements by other people), and the frequency of depressive and anxiety symptoms (fear/panic attacks, sensations of longing, sorrow, tension, etc.), experienced during the last month.

Statistical analysis was carried out using IBM SPSS Statistics 28.0.1 and Statistica V10 Statsoft software. The distribution normalities were challenged with a Shapiro–Wilk test. The normally distributed variables were described by means (M) and standard deviations (SD). The therapy-related dynamics were estimated using paired Student’s t-test for measured variables and χ2 McNemar’s test for dependent proportions; the trends were considered significant at *p* < 0.05. Correlations were estimated using Pearson’s correlation coefficient (for normally distributed data) or Spearman’s rank correlation method (nonparametric). The influences of categorical factors and quantitative variables were estimated by factorial ANOVA and multiple logistic regression methods, respectively.

Ethical approval: the study was approved by Biomedical Ethics Committee at the Kulakov National Medical Research Center of Obstetrics, Gynecology, and Perinatology (Protocol No. 9 of 2020-10-22). All participants (the patients and their legal representatives) provided written informed consent for inclusion in the study, the use of medical data/images, and publication of the findings.

## 3. Results

In the studied sample, the majority of adolescents were characterized with the I stage of endometriosis according to the revised American Society for Reproductive Medicine (rASRM) score (43.8%, 14/32, the rASRM score 3.3 ± 1.2 OR P1-2, O 0/0, T0/0, B1/0 C0 according to #Enzian(s) score). The rest of the patients had the II stage (34.4%, 11/32 the rASRM score 10.8 ± 2.9 or P2, O 0/0, T0/0, B1/2, C0 according to #Enzian(s)) and the III stages of the disease (21.9%, 7/32, the rASRM score 20.4 ± 7.6 or P2-3, B1/2 according to #Enzian(s) score). The III rASRM stage of the disease in two cases was observed on the background of adenomyomas (O2/0), and, in three cases, adhesions on the tubo-ovarian unit (T1/0) were detected.

All patients of the cohort had menarche for the first bleed at the age of 11–15 years. The body mass index (BMI) at admission was 20.6 ± 3.4 kg/m^2^; specifically, <18 kg/m^2^ in 37% (12/32), 18–25 kg/m^2^ in 47% (15/32), and >25 kg/m^2^ in 16% (5/32) of the cohort. Most of the patients had a regular cycle established immediately (9.4%, 3/32) or 1-year post-menarche (46.9%, 15/32), and 44.8% of the cohort (14 patients (pts)) had an irregular cycle. Menstrual abnormalities included a shorter menstrual cycle (<21 days; 15.6%, 5/32), prolonged menses (8–10 days; 6.3%, 2/32), and ‘heavy’ menstrual flow (56.3%, 18/32).

A third part of the patients (28.1%, 9/32) reported non-menstrual vaginal bleeding, including mid-cycle spotting (7 pts) and appearance of discharge 3 days before (4 pts) or 3–5 days before and after each period (2 pts). Among patients with non-menstrual bleeding, 56% (5 pts) had a combined diagnosis of ‘peritoneal EMS/adenomyosis’, and 44% (4 pts) were diagnosed with ‘peritoneal EMS’ only (see Figure 1).

The majority of patients experienced menstrual pain starting from menarche (65.6%, 21/32) or 1.5–2 years post-menarche (21.9%, 7/32). In minor proportions of cases, the pains started 0.5 years post-menarche (2 pts) or were supposedly incited after genital inflammation (1 pt) or after abdominal surgery (1 pt) (see Figure 2).

Most patients (53.1%, 17/32) considered their pain as of constant intensity (in retrospect), whereas in 46.9% (15/32), the intensity increased over time since menarche (see Figure 3a).

When asked to attribute pelvic pain to the course of the menstrual cycle, 34.3% (11/32) of the patients reported mid-cycle pain, and lower proportions reported pain experienced daily (25.0%, 8/32), 1–3 days before menses and that which lasted for several days (25.0%, 8/32) or 3–5 days before and after each period (9.4%, 3/32). In single cases, the pain was experienced a week before or >5 days before and after each period. The structure of pelvic pains as related to the menstrual cycle is shown in Figure 3.

The majority of patients described pain in the lower abdomen ‘above the pubic region’ (40.1%, 13/32), lumbar (21.9%, 7/32), and/or diffuse pains without clear localization (12.5%, 4/32). Other specific sites of painful sensations included the groin area on the left (12.5%, 4/32), groin area on the right (9.4%, 3/32), perineum (2 pts), or sacrum (1 pt; the patients could select more than one answer).

The painful sensations were subjectively specified as ‘dull, nagging’ by 43.8% (14/32) of the patients, whereas 25.0% (8/32) referred to it as ‘spasmodic’. In addition, 21.9% (7/32) had ‘stabbing’ pains, 9.4% (3/32) reported ‘pressing’ pains, and 1 pt defined her sensations as ‘burning’ (the patients could provide more than one definition).

Other clinical signs of EMS included menstruation-related gastrointestinal symptoms (nausea, diarrhea, dyschesia) in 40.6% (13/32) and dysuria in 18.8% (6/32) of the patients. When asked about specific circumstances of pelvic pain worsening (more than one answer allowed), the patients ticked the ‘beginning of menses’ (40.6%, 13/32), ‘physical motion’ (31.3%, 10/32), ‘urination’ (12.5%, 4/32), and ‘during/after defecation’ (9.4%, 3/32). Additionally, 1 pt reported pain worsening after hypothermia or swimming in ponds or pools, 1 pt related it to overheating (bath, sauna, high body temperature), and 2 pts (sexually active) admitted pain worsening during/after sexual intercourse (see Figure 4).

Of note, 62.5% (20/32) of the patients reported having female relatives (mothers, sisters, aunts, maternal/paternal grandmothers) with a history of EMS or abnormal uterine bleedings, and 50.0% (16/32) reported a family history of chronic pelvic/menstrual pain.

The one-year dienogest therapy afforded a substantial reduction in dysmenorrhea pain intensity (as measured by VAS score) from severe (8.3 ± 1.6 points) to mild, approaching physiological levels (1.7 ± 2.1 points) (*p* < 0.001) (Table 1).

The persistence of the pain also showed distinct progestin therapy-related dynamics. Before therapy, 97% (31/32) of the patients experienced pain every menstrual cycle, and only 1 pt reported occasional menstrual pain (≤4 episodes a year). The therapy afforded the cessation of menstrual pain in most of the cases, with the rest of the cohort reporting occasional pain (≤4 episodes a year); in several cases, the pain remained in at least every other cycle (2 pts) or experienced persistent cyclic pain (2 pts) (see Table 1).

The initially high proportion of patients with non-menstrual vaginal bleeding (28%, 9/32) after one-year dienogest therapy was reduced to 12.5% (4/32). Of note, 41% (13/32) of the patients reported a lack of menstrual bleeding during the therapy, which was considered indicative of efficacious therapy.

In addition, the proportion of patients with non-menstrual pelvic pain (60%, 19/32) after one-year dienogest therapy was reduced to 10% (3 pts). The multiple-dose use of painkillers for dysmenorrhea was also considerably reduced (Table 1).

In addition, therapy significantly reduced the incidence of other EMS-associated complaints, including decreased everyday activity/performance and gastrointestinal or dysuria symptoms (Table 1).

The observed alleviation of dysmenorrhea during the treatment was supported by corresponding dynamics not only of VAS but also of McGill Pain Questionnaire scores. One year after the commencement, the test revealed significant decreases in constant/periodic/neuropathic pain and affective pain descriptor scores. These results not only confirm the therapy-related pain alleviation but also indicate a significant contribution of the subjective perception of noxious stimuli (as measured by affective pain descriptors) to the overall severity of EMS-associated pain syndrome (Table 1).

Apart from the reduced severity of EMS-associated pains, the cohort revealed the significant mitigation of affective states/sensations during the therapy (Table 1). Beck’s index, initially corresponding to mild depression (BDI, 10.9 ± 8.9 points) decreased from 3.8 ± 3.8 points corresponding to the absence of clinical signs of depression. The average scores for Spielberger’s ‘reactive’ and ‘inherent’ anxiety subscales (respectively, STAI-S and STAI-T) also decreased significantly, with STAI-T dropping from high to moderate, which reflects the upgrade of physical well-being and probably also a decrease in the tendency of patients to overreact to certain exposures. HADS anxiety and depression subscales also showed a significant reduction at the cohort level, indicating an improvement in the overall psychoemotional status of the patients during the therapy (Table 1).

QoL dynamics for the cohort indicated a significant improvement and approached a state of complete health (100 points) as measured by physical component, psychological component, and average QoL scores on the background of the therapy (Table 1).

As revealed by the interview, the therapy significantly reduced the incidence of heart palpitations and heavy breathing symptoms, possibly regarded as subclinical autonomic neuropathy/dysfunction, at the cohort level. The therapy-related dynamics of fainting episodes and recurrent headaches (reported semi-quantitatively as ‘often’, ‘seldom’, or ‘never’) were negligible. A beneficial trend for recurrent paroxysmal headaches may reflect the overall pain-relieving efficacy of progestins for the studied cohort. Incidentally, the lack of significant dynamics for headaches, which may arise as a side effect of progestin intake, further justifies the continuous medication protocol. In the study sample, one patient noted an increase in body weight (3.1%), four patients noted a decrease (12.5%) compared to the initial BMI, and the rest of the girls did not notice a change in weight during dienogest therapy. The lack of significant dynamics for BMI during treatment, which is especially important for adolescents, also indicates good tolerability and contributes to high satisfaction with the treatment reported by participants.

In addition, the decrease in cognitive/affective signs, including mild memory difficulties, explosiveness (uncontrollable nervous flashes), nervousness/tension/anxiety, and fear/panic attacks, was also revealed by the interview (see Table 2). In addition, the patients reported a reduction in subjectively perceived negative emotional experiences (senses of guilt, anguish and sadness, depression, worthlessness, loneliness, alarm, and overall self-dissatisfaction) during the therapy.

Peripheral blood tests revealed a decrease in testosterone, estradiol, prolactin, cortisol, and CA-125 levels, and neutrophil-lymphocyte count ratio during the one-year dienogest therapy (all within reference ranges) amid an increase in LH and FSH levels (Table 3).

A multifactor regression analysis involving objective and subjective parameters was carried out to identify the predictors of affective signs among clinical and laboratory indicators and ultimately to assess the efficiency of dienogeat therapy in young patients with EMS. According to the results, elevated estradiol levels represented a significant independent predictor of high HADS anxiety (F = 5.62, *p* = 0.004), HADS depression (F = 9.86, *p* = 0.033), and Spielberger’s ‘reactive’ anxiety (STAI-S; F = 3.16, *p* = 0.046) scores, as well as the McGill affective pain descriptor and constant pain scores (respectively, F = 9.79, *p* = 0.004 and F = 12.49, *p* = 0.001). In addition, lower BMI and higher CRP were identified as significant independent factors of a high VAS score (respectively, F = 7.71, *p* = 0.009 and F = 4.18, *p* = 0.049), whereas FSH levels significantly influenced physical well-being as assessed by SF-36 (F = 5.29, *p* = 0.028).

Another aspect was the interaction of subjective factors with affective signs persistence. Significant predictors of high BDI scores included low scores for physical and psychological QoL components (respectively, F = 7.13, *p* = 0.012 and F = 14.39, *p* < 0.001), as well as the averaged QoL score (F = 13.88, *p* < 0.001), as assessed by SF-36. Apart from BDI, the QoL psychological component was negatively influenced by pain intensity (VAS; F = 4.60, *p* = 0.040), HADS anxiety (F = 5.60, *p* = 0.025), and HADS depression (F = 4.85, *p* = 0.036) scores. The QoL physical component was negatively influenced by pain syndrome indicators, including VAS (F = 16.00, *p* < 0.001) and McGill’s constant (F = 12.84, *p* = 0.001), neuropathic (F = 7.49, *p* = 0.010) and affective pain descriptor (F = 7.12, *p* = 0.012) scores.

It is important to highlight the prognostically meaningful connection between levels of certain hormones before the therapy and the degree of anxiety-depression signs even one-year post-commencement, notably the influence of initial estradiol levels on final scores for STAI-S (F = 9.45, *p* = 0.004), STAI-T (F = 5.65, *p* = 0.024) and HADS anxiety (F = 7.97, *p* = 0.008) subscales, respectively.

## 4. Discussion

EMS manifestations, including severe dysmenorrhea, chronic pelvic pain, and intermenstrual spotting, pose a considerable threat to mental and somatic health in young women [16,17,18,19,20,21,22,23,24,25]. In most cases, the pain is experienced starting from menarche and is not necessarily confined to menses [32,33]. With the estimated average diagnostic delay of 7–10 years especially pronounced in young patients, such pain often leads to hyperalgesia, central sensitization, and mood symptoms (anxiety, exhaustion, etc.), forming a vicious circle with EMS progression [34]. This situation was reflected in a number of studies demonstrating the negative impact of EMS on mental health in patients and their partners [35]. Such studies, however, seldom involve adolescents; meanwhile, the early-manifested disease can take a highly aggressive course ensuing mental complications [33].

The varying localization of EMS-associated pains, often atypical (diffuse pain of unclear localization; in the groin, perineum, or sacrum) with pain sensations varying in character (baking, cutting, stabbing, pressing, etc.), indicate the presence of a neuropathic component. The variations can be explained by the complex topography of immune dysregulation and shifted macrophage equilibria, as well as the estrogen-induced innervation of the lesions. The estrogen-dependent inflammation in EMS foci triggered multiple mechanisms of sensory and sympathetic neurogenesis amid the gradual aggravation of the local pro-inflammatory status. The persistent cytokine-induced stimulation of nociceptors facilitates neuroinflammation through the release of inflammatory neurotransmitters. The results of conducted multivariate analysis linked CRP levels to the VAS score and estradiol levels to the McGill pain index. Our data fit into the conception that estrogen-induced neuro-immune interaction plays a key role in peripheral sensitization while enhancing EMS-associated neuropathic pains [24,25,26].

Correlations between BMI and an affective component in EMS are of special clinical interest. Interaction between these parameters is revealed by factorial analysis (low BMI, higher VAS score; *p* = 0.009). This finding is consistent with the reported positive influence of severe dietary restrictions combined with high physical activity (typical of anorexia nervosa) on EMS progression by Reis F.M., Coutinho L.M. et al., 2020 [36]. At the same time, EMS itself can promote a decrease in body weight through impaired liver metabolism, as demonstrated by Goetz L.G. et al. in the mouse model: animals with experimentally induced EMS showed a higher expression of four presumably ‘anorexigenic’ genes and lower expression of two genes associated with obesity [37].

EMS can be treated either hormonally or surgically, with laparoscopy remaining the gold standard for its diagnostics in all susceptible age groups. Progestins-based regimens are considered a first-line pathogenetic option in EMS, with minimum risks and maximum clinical benefits [4]. A study on dienogest efficacy in 12–18-year-olds with clinically suspected EMS by Eber et al. revealed the significant mitigation of pain syndrome [4]. In our setting, the continuous administration of dienogest for 1 year in combination with NSAIDs for symptomatic pain relief after surgical treatment showed high efficacy in reducing EMS-associated pains and concomitant symptoms. The positive dynamics were indicated by a decrease in pain intensity (VAS score, *p* < 0.001), reduced intermenstrual pain incidence (*p* < 0.001), and reduced NSAID intake (*p* < 0.001). The achieved therapeutic effect apparently involved the COX-dependent suppression of prostaglandin synthesis by NSAIDs, prolonged and complemented by the anti-inflammatory action of, which are known to inhibit the production of pro-inflammatory TNF-α, IFN-γ, and IL-12 and activate the synthesis of anti-inflammatory IL-10 in EMS foci. In addition, progestins are known to reduce the pro-inflammatory impact of antigen-presenting cells, notably dendritic cells, while supporting the regeneration and functional maturation of epithelial lining in the uterus [38].

The therapy also alleviated affective symptoms as measured by BDI, including Spielberger’s reactive/inherent anxiety subscale scores and HADS anxiety and depression scores (*p* = 0.000 for each). Before the therapy, patients experienced moderate reactive/high inherent anxiety (Spielberger’s) and sub-depression (BDI). The reduction in scores to normative values indicated a substantial improvement in the psycho-emotional status of the patients. Apparently, this recorded decrease in pain intensity eventually alleviated the distress associated with the expectation and sensation of cyclic and non-cyclic pains, annoying (sometimes to the point of exhaustion) and interfering with everyday activities [8]. This assumption was supported by significant beneficial dynamics in autonomic responses characteristic of mental disorders, including heart palpitations (↓, *p* = 0.001) and heavy breathing (↓, *p* = 0.000), as well as neurotic signs (‘flashes’, ↓, *p* = 0.003; ‘nervousness’, ↓, *p* = 0.000; ‘fear/panic attacks’, ↓, *p* = 0.000) and cognitive impairment (‘memory difficulties’, ↓, *p* = 0.001). Our data support a close relationship between the degree of EMS-associated pains and anxiety/depression signs originating in adolescence [10,14].

Furthermore, the study revealed a significant improvement in life quality indicators during therapy, including ‘physical’, ‘psychological’, and ‘overall life quality’ domains (*p* = 0.000 each). These therapy-related positive dynamics can be explained by the alleviation of both physical symptoms and mental complications (depression, anxiety) of the disease. The alleviation of physical symptoms is indicated by the significantly reduced proportion of patients reporting hampered everyday activity, weakness, and declined capabilities during menses, gastrointestinal symptoms, and dysuria after the therapy. Our data are consistent with the results of Mińko A. et al., relating to a pronounced decrease in the quality of life and higher rates of depression and anxiety scores in patients with endometriosis [39], as well as the pronounced relief of symptoms during treatment in adults [40]. We suppose it is important to underline that adolescents also experience impairment in QoL, similar to adults already in their childhood, which reiterates the need for the timely initiation of pathogenetic therapy [34].

Significant decreases in testosterone, prolactin, estradiol, and cortisol levels, along with the opposite dynamics for LH and FSH (all within reference ranges), indicate that this therapy does not induce hypothalamic-pituitary-ovarian/adrenal axis destabilization. In addition, we observed a significant decrease in the neutrophil-to-lymphocyte count ratio and CA-125 levels, accompanied by a similar trend for CRP; these dynamics may also indicate the alleviation of inflammatory processes by continued therapy.

The search for predictors of affective disorders among clinical and laboratory parameters identified high estradiol levels as an independent significant positive factor of HADS anxiety, HADS depression, and Spielberger’s reactive anxiety scores (*p* = 0.004, *p* = 0.033 and *p* = 0.046, respectively). Still, published evidence on the influence of estrogens on psycho-emotional status is controversial. It is conventionally maintained that estrogens boost noradrenalin and β-endorphin synthesis while inhibiting monoamine oxidases and modulating dopamine receptor activities in brain tissue, thereby promoting anxiolytic and anti-depressant effects [41]. The mechanisms also involve a boost in the number of active sites for serotonin intake by the brain cells and increased production of acetylcholine through catechol-O-methyltransferase stimulation. At the same time, one study showed that estradiol supported the expression of tryptophan hydroxylase 2, converting tryptophan to serotonin while simultaneously inhibiting the expression of serotonin receptor 1A and monoamine oxidases A and B at the transcriptional level [42]. Therefore, the onset of affective symptoms may reflect a shift in serotonin equilibrium backed by high estradiol levels. Specifically, the inflammatory conditions-favored depletion of serotonin through conversion to 5-hydroxykynurenine upon indolamine-2,3-dioxygenase activation, along with the lack of accessory serotonin receptors (e.g., A2) may significantly contribute to affective components in EMS [43].

The comparative assessment of psycho-emotional factors in the structure of the affective component identified low SF-36 scores in physical, psychological, and average QoL (*p* < 0.05 for each) domains as significant predictors of depression (as assessed by BDI). Apart from the BDI score, life quality indicators negatively correlated with the VAS score for pain intensity and HADS scores for anxiety and depression (*p* < 0.05 for each). In addition, subjective indicators of psycho-emotional status and life quality were interrelated: for instance, psychological well-being significantly reflected HADS scores and BDI, while physical well-being reflected McGill scores for constant/neuropathic pains and affective pain descriptors, and both psychological and physical domain scores significantly reflected the VAS score. These results are consistent with the supposedly central role of pain syndromes in an EMS-associated sense of physical and psychological deprivation [3,33].

The search for biochemical predictors of EMS symptom severity identified CRP levels as an independent significant correlate of the VAS score (*p* = 0.05). Despite the lack of significant therapy-related dynamics for the studied cohort, CRP levels could be used to estimate the grade of inflammatory process on an individual basis.

Estradiol levels significantly influenced McGill scores for constant pain and affective pain descriptors (*p* < 0.005), which can reflect the increased production of estradiol by active EMS foci and stress responses to persistent noxious stimuli leading to central and peripheral sensitization. No specific relationships between hormone levels and psycho-emotional characteristics in adolescents with EMS have been described so far [4,44]. Considering a plausible connection between low BMI with EMS in terms of risks and severity, the elevated systemic estradiol levels may independently correlate with anorexic behaviors ranging from borderline to severe. The identified objective and subjective predictors of affective complications in adolescents with EMS can be validated for clinical use to enable the timely correction and improvement of the therapy in order to optimize the outcome and preserve both the somatic and psycho-emotional health of the patients.

The shortcomings of our study include the small size of the cohort and non-comparative observational design (i.e., without a control group(s) enrolling matched conditionally healthy individuals and/or patients receiving alternative treatment or non-receiving progestins after surgery treatment).

Thus, adolescent patients with genital EMS are likely to develop characteristic symptoms of mood disorders; importantly, this tendency was revealed by different questionnaires. The collected data indicate high-stress susceptibility and the aggravation of affective symptoms/neuroticism in adolescents with a distinctive clinical picture of EMS, especially against the background of non-effective symptomatic treatment.

It should be noted that EMS-associated chronic pain management in adolescents is often more complicated than in adults, as the therapist has to explicitly communicate with both the patient and her parents. In addition, every clinical decision must account for the young age of the patient to ensure her long-term emotional/physical/sexual well-being and fertility. On one hand, persistent pelvic pains can promote mood disorders (depression, anxiety) or aggravate them. On the other hand, adolescents with signs of depression/anxiety may have an exaggerated perception of painful experiences and are more likely to develop inadequate coping strategies, e.g., social isolation [6]. The continuous physical and emotional maturation status of the patient requires complex personalized approaches; ultimately, EMS should be handled as an essential developmental condition at all stages of clinical management, including history collection, examination, and treatment.

It is important to emphasize that the elimination of clinical signs in EMS does not ensure the compensation of a serotonin system deficit. In this regard, young patients with EMS-associated chronic pelvic pain may substantively benefit from pathogenetically justified psychotropic medication regimens [25].

Considering the multiple systemic effects of EMS, the comprehensive pain-focused treatment schemes accounting properly for psychological consequences are advantageous. To provide maximum relief, the management should be accomplished through coordinated efforts of gynecologists, psychiatrists, psychologists, and physiotherapists. A solid body of research evidence identifies juvenile EMS as a key medical and social issue requiring advanced diagnostics and management with a mandatory account for psycho-emotional and life-quality indicators.

## 5. Conclusions

After one-year dienogest treatment, the patients reported significant alleviation of the severity of dysmenorrhea and EMS-associated symptoms, including pain-related decreases in daily activities and performance; gastrointestinal/dysuric symptoms (*p* < 0.001 for each). The therapy afforded significant relief of affective symptoms (BDI, HADS anxiety/depression, and STAI reactive/inherent anxiety scores) and quality-of-life improvement (SF-36 QoL scores; *p* < 0.005);

Patients with endometriosis reported high satisfaction with one-year dienogest therapy: no side effects (significant BMI changes, headaches) were encountered, although 13% of the adolescents reported episodes of spotting. The therapy promoted a decrease in blood levels of hormones and inflammation markers (prolactin, estradiol, cortisol, testosterone, neutrophil/lymphocyte ratio, CA-125) within reference ranges.

Estradiol levels provide a significant predictor for affective disorders in adolescents with endometriosis before treatment (STAI reactive anxiety; HADS anxiety/depression; McGill affective/constant pain score) and after the therapy (STAI and HADS anxiety scores), whereas BMI and CRP levels are, respectively, negative and positive factors of pain syndrome severity.

## Figures and Tables

**Figure 1 jcm-12-02400-f001:**
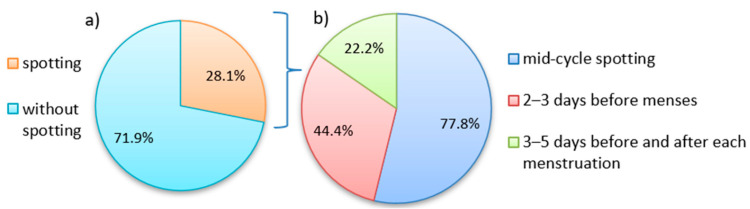
The structure of non-menstrual spotting in the study’s sample: (**a**) the frequency of spotting in the sample, (**b**) the structure of spotting in relation to menstrual cycle.

**Figure 2 jcm-12-02400-f002:**
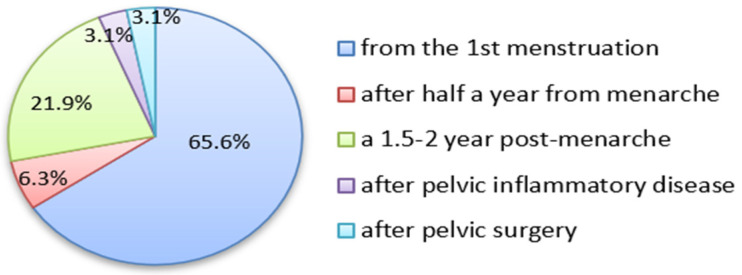
The structure of dysmenorrhea/pelvic pain onset since menarche.

**Figure 3 jcm-12-02400-f003:**
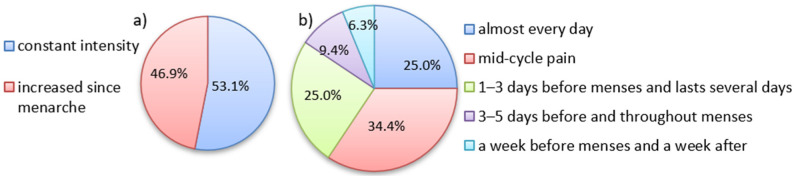
The structure of pelvic pain: (**a**) The dynamics of pain intensity since menarche, (**b**) Pattern of pelvic pain onset in relation to menstrual cycle.

**Figure 4 jcm-12-02400-f004:**
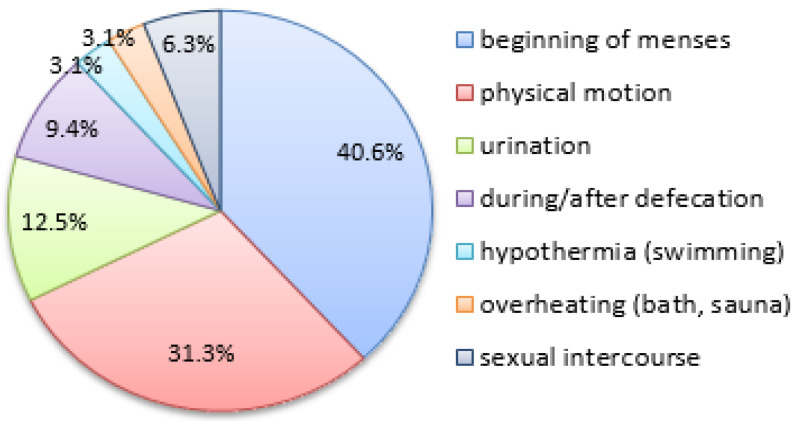
The factors of pelvic pain worsening in the study’s sample.

**Table 1 jcm-12-02400-t001:** Pain, anxiety, depression, and quality-of-life indicators before and after one-year dienogest-therapy.

Parameters	Before Therapy, % (*n* = 32)	After Therapy, % (*n* = 32)	*p*-Level
VAS score (severity of dysmenorrhea) *	8.3 ± 1.6	1.7 ± 2.1	<0.001
Persistent to NSAIDs dysmenorrhea *#*	96.9% (31)	6.3% (2)	<0.001
Chronic non-cyclical pelvic pain *#*	59.4% (19)	9.4% (3)	0.107
Desmenorrhea ≥ 6 episodes a year *#*	0	6.3% (2)	0.0157
Desmenorrhea ≤ 4 episodes a year *#*	3.1% (3)	28.1% (9)	0.011
No pelvic pain *#*	0%	59.4% (19)	<0.001
The need for repeated NSAID administration during menses *#*	75.0% (24)	12.5% (4)	<0.001
Restricted everyday activity/performance *#*	75.0% (24)	6.3% (2)	<0.001
Weakness and decreased capabilities during menses *#*	71.9% (23)	6.3% (2)	<0.001
Gastrointestinal symptoms (nausea, diarrhea, dyschesia) *#*	40.6% (30)	6.3% (2)	0.001
Dysuria *#*	21.8% (7)	0	0.008
SF-MPQ-2*			
Constant pain Periodic pain Neuropathic pain Affective descriptors	3.4 ± 1.7 2.3 ± 1.7 1.6 ± 1.3 5.0 ± 2.8	0.5 ± 0.8 0.2 ± 0.5 0.1 ± 0.3 0.5± 0.9	<0.001 <0.001 <0.001 <0.001
BDI *	10.9 ± 8.9	3.8 ± 3.8	<0.001
HADS anxiety *	6.8 ± 3.2	3.8 ± 2.1	<0.001
HADS depression *	4.5 ± 3.2	1.9 ± 1.9	<0.001
Inherent anxiety (STAI-T) *	47.8 ± 11.4	37.4 ± 9.9	<0.001
Reactive anxiety (STAI-S) *	42.9 ± 10.6	33.8 ± 8.3	<0.001
Quality-of-life questionnaire SF-36 *			
Physical component Psychological component Averaged score	54.1 ± 20.7 52.3 ± 20.7 53.2 ± 18.6	79.9 ± 11.2 71.0 ± 9.6 75.5 ± 9.8	<0.001 <0.001 <0.001

* Data presented as mean ± standard deviation; Student’s *t*-test for dependent samples. *#* data presented as % (absolute); χ^2^ McNemar’s test for dependent proportions. VAS, Visual Analog Scale pain scoring; SF-MPQ-2, Short-Form McGill Pain Questionnaire 2; BDI, Beck Depression Index; HADS, Hospital Anxiety and Depression Scale; STAI, Spielberger State-Trait Anxiety Inventory (respectively, STAI-S and STAI-T subscales); SF-36, 36-Item Short Form Survey quality-of-life scoring.

**Table 2 jcm-12-02400-t002:** Cognitive, affective symptoms and negative emotional sensations before and after dienogest therapy.

Parameter (“How Often in the Last Month Have You Experienced …?”)	Before Therapy, % (*n* = 32)	After Therapy, % (*n* = 32)	*p*-Value
Heart palpitations/heart arrest *#* often seldom never	9.4% (3) 37.5% (12) 53.1% (17)	0 12.5% (4) 87.5% (28)	0.080 0.023 0.003
Heavy breathing/shortness of breath/rapid breathing *#* often seldom never	3.1% (1) 53.1% (17) 43.8% (14)	0 3.1% (1) 96.9% (31)	0.316 <0.001 <0.001
Paroxysmal headaches *#* often seldom never	12.5% (4) 62.5% (20) 25.0% (8)	0 75.0% (24) 25.0% (8)	0.039 0.281 1
Fainting episodes *#* often seldom never	3.1% (1) 56.3% (18) 40.6% (13)	0 59.4% (19) 40.6% (13)	0.316 0.802 1
Mild memory difficulties *#* none mild moderate considerable serious	50.0% (16) 3.1% (1) 25.0% (8) 9.4% (3) 12.5% (4)	71.9% (23) 25.0% (8) 3.1% (1) 0 0	0.073 0.012 0.012 0.076 0.039
Explosiveness (nervous flashes, that you cannot control) *#* none mild moderate considerable	43.8% (14) 25.0% (8) 9.4% (3) 21.9% (7)	62.5% (20) 31.3% (10) 6.3% (2) 0	0.134 0.575 0.645 0.005
Anxiety, tension *#* never almost never sometimes rather often typically	6.3% (2) 21.9% (7) 28.1% (9) 31.3% (10) 12.5% (4)	18.8% (6) 43.8% (14) 37.5% (12) 0 0	0.131 0.062 0.423 0.006 0.039
Fear/panic attacks *#* never almost never sometimes rather often typically	15.6% (5) 34.4% (11) 25.0% (8) 21.9% (7) 3.1% (1)	34.4% (11) 53.1% (17) 12.5% (4) 0 0	0.003 0.132 0.200 0.005 0.316

# data presented as % (absolute); χ^2^ McNemar’s test for dependent proportions.

**Table 3 jcm-12-02400-t003:** Blood levels of hormones and inflammation markers before and after dienogest therapy.

Parameters	Before the Therapy	After Therapy	*p*-Value
LH *, IU/L	6.75 ± 3.08	8.22 ± 2.37	<0.001
FSH *, IU/L	5.39 ± 2.04	6.63 ± 2.01	<0.001
Prolactin * mIU/L	528.14 ± 302.92	323.50 ± 191.50	<0.001
Estradiol *, pmol/l	250.80 ± 200.53	91.35 ± 68.51	<0.001
Testosterone *, nmol/L	1.00 ± 0.36	0.82 ± 0.37	0.010
Cortisol *, nmol/L	401.27 ± 148.27	268.31 ± 118.85	<0.001
Neutrophils/Lymphocytes	2.49 ± 1.20	1.56 ± 0.68	<0.001
CA-125 *, U/mL	26.74 ± 21.56	12.05 ± 5.82	<0.001
CRP *, mg/L	1.22 ± 1.46	0.75 ± 0.43	0.059

* Data presented as mean ± standard deviation; Student’s *t*-test for dependent samples. LH, luteinizing hormone; FSH, follicle-stimulating hormone; CRP, high-sensitivity C-reactive protein test.

## Data Availability

The data presented in this study are available upon request from the corresponding author.

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
