# Peer review of "Pelvic Pain, Mental Health and Quality of Life in Adolescents with Endometriosis after Surgery and Dienogest Treatment"

_jcm, 2023, doi:10.3390/jcm12062400_

Round 1
Reviewer 1 Report
Well done to all authors for investigating such an under- researched area!
improvements (numbers refer to lines)
21- researches this is only ever used as singular: research
106 diagnosis of peritoneal EMS confirmed by instrumental examination (ultrasound, MRI) and laparoscopy - the underlined words do not read well- I suggest say n= x had ultrasound/MRI 9state criteria, e.g. Enzian or ultrasound - which diagnostic criteria used) laparoscopy n=x and state if lesions were histologically confirmed.
also - please specify - if the patients had a laparoscopy were the lesions removed? This could influence the response to treatment
materials and methods - how did you decide on the number of participants , did you do a power study, were they sampled convenience and were they consecutive patients of somehow selected - was recruitment prospective or retrospective
did the girls/parents agree to participate and have their data collected (ethics)
137 “Psychodiagnostics” - the pain score does not first under this heading
208-220 there is no plural vaginal bleedings it should say vaginal bleeding/ menarche for first bleed
247 - pain in low abdomen ‘above the lap’ - reword - “pain in the lower abdomen” above the lap- this does not make sense in the English language - do the authors refer to pelvis or pubic region?
314 pain - replace with pain - it is usually not used as a plural
results section - cut by 1/3 and refer to tables rather than spelling all out in text - just highlight the most important findings, otherwise the reader gets a but overwhelmed by data description.
discussion is too long - cut by 50% just highlighting the therapeutic effects of the intervention
conclusion is very good and succinct
I wish the authors good luck with their publication!
Author Response
We are sincerely grateful to the reviewer for the constructive thorough criticism and interest in our study. The point-by-point answers to the comments are given below.
Point 1. 21- researches this is only ever used as singular: research
Response 1. Thank you very much for your help, we re-read the text ant replaced (lines 21, 57, 70).
Point 2. 106 diagnosis of peritoneal EMS confirmed by instrumental examination (ultrasound, MRI) and laparoscopy - the underlined words do not read well - I suggest say n= x had ultrasound/MRI 9state criteria, e.g. Enzian or ultrasound - which diagnostic criteria used) laparoscopy n=x and state if lesions were histologically confirmed.
also - please specify - if the patients had a laparoscopy were the lesions removed? This could influence the response to treatment
Response 2. Thank you for this very important clarification.
In this paper we included only patients with already confirmed peritoneal endometriosis according to laparoscopy and histology.
We changed:
Line 106: diagnosis of peritoneal EMS was laparoscopically and histologically confirmed after instrumental examination (ultrasound, MRI).
We detailed the indications for laparoscopy:
Line 108-112. The indications for laparoscopy were: the unsuccessful empirical treatment (NSAIDs, antispasmodics, COCs) or the persistent moderate/severe dysmenorrhea with negative imaging results. Data from the laparoscopic picture included surgical diagnosis, the stage of endometriosis according to the revised American Society for Reproductive Medicine (rASRM) criteria. In each case of laparoscopy, we preferred excision of peritoneal foci, coagulation was performed in less accessible places.
We added in the Results the rASRM scores and stage of the disease in our sample:
Line 216: In the studied sample the majority of adolescents were characterized with the I stage of endometriosis according to revised American Society for Reproductive Medicine (rASRM) score (43.8%, 14/32, the rASRM score 3.3±1.2) and the rest of patients - with the II stage (34.4%, 11/32, the rASRM score 10.8±2.9) and the III stage of the disease (21.9%, 7/32, the rASRM score 20.4±7.6).
Understanding the significance of this comment and influence on the results and response to treatment, we changed the title of the article:
Pelvic pain, mental health and quality of life in adolescents with endometriosis after surgery and dienogest treatment
We do have difficulties in diagnosis of initial forms of peritoneal EMS in adolescents according to imaging data. We summarized our clinical results in previous paper on instrumental diagnosis of peritoneal endometriosis in adolescents with ultrasound and MRI compared to laparoscopy (https://www.mdpi.com/2077-0383/12/4/1678). According to our results, it was possible to detect initial stages of peritoneal EMS in 3.3% cases by ultrasound and in 78.9% by MRI, likely because in adolescents we see early superficial forms of the disease, and extremely rarely - an infiltrative process involving adjacent organs.
Before laparoscopy, we performed ultrasound and MRI to all patients with suspected peritoneal endometriosis. Further, if symptomatic therapy was insufficient and moderate-severe dysmenorrhea remained even in the absence of MRI suspicion of peritoneal endometriosis, we performed laparoscopy with diagnostic and therapeutic purposes. After confirming the diagnosis, we prescribe progestins in a prolonged regimen to prevent the progression and recurrence of the disease, while maintaining quality of life.
In the attachment we added some photos as an example to emphasize the type of lesions we usually see at the laparoscopy in adolescents. Given the superficial lesions in the most of cases and small sizes of foci (less than 3-5 mm in average), we really experience difficulties in diagnosing such lesions according to ultrasound and MRI.
Point 3. Materials and methods - how did you decide on the number of participants , did you do a power study, were they sampled convenience and were they consecutive patients of somehow selected - was recruitment prospective or retrospective
did the girls/parents agree to participate and have their data collected (ethics)
Response 3: Thank you for this comment.
We calculated sample size using power analysis of the Statistica 12 software. The minimal enough sample size was from 5 to 20 girls in one group. We based on results of dependent-sample T-test (calculated according to the literature date on adolescents with endometriosis during dienogest therapy according to dynamics of Vas-score (from 64.3 mm (SD, 19.1 mm) to of 9.0 mm (SD, 13.9)) and McNemar’s Test for two-proportions paired samples on frequency of EMS related symptoms (Ebert et al., 2017).
[Ebert, A. D., Dong, L., Merz, M., Kirsch, B., Francuski, M., Böttcher, B., Roman, H., Suvitie, P., Hlavackova, O., Gude, K., & Seitz, C. (2017). Dienogest 2 mg Daily in the Treatment of Adolescents with Clinically Suspected Endometriosis: The VISanne Study to Assess Safety in ADOlescents. Journal of Pediatric and Adolescent Gynecology, 30(5). https://doi.org/10.1016/j.jpag.2017.01.014].
In order to obtain results on a sample with confirmed peritoneal endometriosis we conducted a retrospective study:
Line 117: Study design: retrospective longitudinal observational cohort study.
Thank you for this comment. We placed the Ethical approvement of the study as it is in a template of the paper:
Line 575:
Institutional Review Board Statement: The study was conducted in accordance with the Declaration of Helsinki, and the protocol was approved by the Ethics Committee for Biomedical Research at the National Medical Research Center for Obstetrics, Gynecology and Perinatology named after Academician V.I. Kulakov Ministry of Healthcare of the Russian Federation, Moscow, Russia (Project identification code â„– 9, 22 October 2020).
Informed Consent Statement: Informed consent was obtained from all subjects (or their legal representatives in cases where the girls were under 15 years old) involved in the study. Written informed consent has been obtained from the patient(s) to publish the paper.
Point 4. 137 “Psychodiagnostics” - the pain score does not first under this heading
Response 4: Thank you for this correction. We clarified:
Line 143: The tests and interview included.
Point 5. 208-220 there is no plural vaginal bleedings it should say vaginal bleeding/ menarche for first bleed
Response 5: Thank you very much, we corrected “menarche for first bleed” (line 214) and “vaginal bleeding” (line 221-223).
Point 6. 247 - pain in low abdomen ‘above the lap’ - reword - “pain in the lower abdomen” above the lap- this does not make sense in the English language - do the authors refer to pelvis or pubic region?
Response 6: Thank you a lot. We meant pubic region. We clarified:
Line 252 - lower abdomen ‘above the pubic region’
Point 7. discussion is too long - cut by 50% just highlighting the therapeutic effects of the intervention.
conclusion is very good and succinct
Response 7: Thank you very much for your attention and clarification. We have tried to shorten the discussion by focusing on the topic and deleting all statistical data from the section. At the same time, we think it is very important to analyze and explain hypothetically why we are seeing such changes and the results, given the lack of research on this topic in adolescents.
We thank You very much for your valuable comments, critics and time to improve our paper!
Please see the attachment below!

Author Response
Response to Reviewer Comments
Esteemed Reviewer!
We are sincerely grateful to thorough criticism to our study. We did our best to meet the requirements. The point-by-point answers to the comments are given below.
Point 1. Line 73: please avoid only necessary information.
Response 2. Thank you, we deleted this part (lines 73-76).
Point 2. In how many patients did you confirm by laparoscopy. Did you have any findings in TVS or MRI? Did you use a classification system? Could you provide the score? During which period of time did you confirm these 32 patients?
Response 2. Thank you for this important comment.
We included only patients with already confirmed peritoneal endometriosis according to laparoscopy and histology. The laparoscopy was performed for diagnostic and therapeutic purposes. In each case the visible endometriotic foci were removed. After the surgery dienogest was prescribed and the results were analyzed after a year treatment in comparison with the data before operation was performed.
We conducted a study after ethical approvement in 2020, in 2022 we analyzed the results, obtained after a year therapy.
We find it very important to summarize the data on MRI and ultrasound compared to laparoscopy more in detail in our previous paper (https://www.mdpi.com/2077-0383/12/4/1678). According to our results (obtained on 90 patients with peritoneal endometriosis), it was possible to detect initial stages of peritoneal EMS in 3.3% cases by transabdominal ultrasound and in 78.9% by MRI, likely because in adolescents we see mostly superficial forms of the disease, and rarely - an infiltrative process. We described on a larger sample the most essential imaging signs, that could be helpful in early EMS evaluation in adolescents.
Each patient after a surgery receives progestin therapy in order to prevent recurrence, reduce complaints if remain and maintain the quality of life. We plan to summarize the results for the entire sample of patients in more detail in the future when we have these data. At the same time, we do observe very positive changes in the clinical characteristics of patients, a pronounced decrease in complaints and a significant improvement in the quality of life of patients with the recovery of daily activity and performance (social, school, emotional, etc., which is very important for young girls) during therapy, so we wanted to share our results.
We changed:
Line 106: diagnosis of peritoneal EMS was laparoscopically and histologically confirmed after instrumental examination.
We detailed the indications for laparoscopy:
Line 108-112. The indications for laparoscopy were: the unsuccessful empirical treatment (NSAIDs, antispasmodics, COCs) or the persistent moderate/severe dysmenorrhea with negative imaging results. Data from the laparoscopic picture included surgical diagnosis, the stage of endometriosis according to the revised American Society for Reproductive Medicine (rASRM) criteria. In each case of laparoscopy, we preferred excision of peritoneal foci, coagulation was performed in less accessible places.
We added in the Results the rASRM scores and stage of the disease in our sample:
Line 216: In the studied sample the majority of adolescents were characterized with the I stage of endometriosis according to revised American Society for Reproductive Medicine (rASRM) score (43.8%, 14/32, the rASRM score 3.3±1.2) and the rest of patients - with the II stage (34.4%, 11/32, the rASRM score 10.8±2.9) and the III stage of the disease (21.9%, 7/32, the rASRM score 20.4±7.6).
Understanding the significance of this comment and influence on the surgery on the results and response to treatment, we changed the title of the article:
Pelvic pain, mental health and quality of life in adolescents with endometriosis after surgery and dienogest treatment
Point 3. Line 219: How did you diagnose adenomyosis? Please describe the criteria.
Response 3: Thank you for this comment.
We added in the text, that we considered imaging data in diagnosing adenomyosis.
Line 101-102: Besides, 43,8% (14/32) girls have association of peritoneal endometriosis with diffuse adenomyosis according to imaging data (US and MRI).
In our everyday work the diagnosis of adenomyosis we established on the basis of imaging data. According to transabdominal ultrasound such signs as: uneven contour and serrated edge of the endometrium, heterogeneity of the transition zone, uneven echogenicity of the myometrium and asymmetric thickness of the anterior and posterior walls of the uterus are usually described (US transabdominal detection of adenomyosis according to our data is 30-40% in comparison to MRI).
According to MRI the most common signs of adenomyosis in adolescents in the study sample (14 girls) were similar to those, described for adults: asymmetric thickness of the uterine wall (78.5%; 11/14), decrease in zonal differentiation of the uterus (50.0%; 7/14), heterogeneous structure of the myometrium with increased MR signal (85.7%; 12/14), heterogeneous or hypointense MR signal of the endometrium (100%; 14/14), uneven thickening of the transition zone (57.1%; 8/14), uneven contours of the transition zone (92.9%; 13/14), heterogeneous structure of the transition zone (78.5%; 11/14).
Point 4. Line 225: Do you think that the pain of two patients was related to inflammation and surgery, or endometriosis?
Response 4: Thank you for this comment. Because we include in the study only patients with confirmed diagnosis of endometriosis, we consider this pain related to EMS. Surgery and inflammation the girls noted in the medical history as anamnestic factor.
Point 5. Did you experience any bleeding disorders under treatment in the group of patients with adenomyosis?
Response 5: Indeed, we analyzed bleeding disorders during therapy. And these disorders were more likely in the group of patients with adenomyosis. Exactly:
Line 304. Initially high proportion of patients with non-menstrual vaginal bleeding (28%, 9/32) after one-year dienogest therapy reduced to 12.5% (4/32).
Thus, in a small percentage of cases, patients were characterized by prolonged spotting under therapy, which did not lead to anemia and was not an indication for discontinuation of therapy in any case. However, these patients required more frequent outpatient visits and, if necessary, hemostatic therapy with tranexamic acid.
Unfortunately, we have not enough sample yet to perform an analysis of the follow-up results of longer-term progestin treatment in adolescents. In a few cases (two patients) such long-term spotting became less frequent and disappeared after several years of therapy (two and three years).
We are continuing to follow up patients on progestin treatment, including bone mineral density and estrogen levels, and would be happy to provide these data in the future.
Point 6. In my personal experience, treatment with dienogest is linked to several possible side-effects, e.g. depression, anxiety, higher BMI, etc.
How do you explain that you have such a positive result in all parameters in your study cohort? Please discuss.
Response 6: Thank you for this fair note. The focus of our attention was to study the side effects of progestins therapy, since very often patients and their parents are afraid of taking hormonal drugs at this young age. We are accumulating evidence of no significant effect of dienogest on BMI, at least in adolescents.
We added in the results to highlight the lack of effect on BMI:
Line 350: The lack of significant dynamics for BMI during treatment, which is especially important for adolescents, also indicates good tolerability and contributes to high satisfaction with the treatment reported by participants. In the study sample, one patient noted an increase in body weight (3.1%), 4 patients noted a decrease (12.5%) comparing to the initial BMI, the rest of the girls did not notice a change in weight.
Concerning depression and anxiety, indeed, patients with endometriosis with detailed interviewing were characterized with various symptoms, partly psychosomatic, partly due to persistent high level of recurrent pelvic pain. For us, as pediatric gynecologists, it turned out to be quite unexpected such a spread of complaints on negative emotional sensations and decrease in the quality of life in adolescents. Probably in this regard, the relief of pain symptoms was associated with such a positive dynamics on the scales of depression and anxiety in adolescents. It should be noted that it is possible that the longer the patient experiences chronic pain syndrome, the worse the dynamics may be during treatment, including in connection with the anxious accentuation of the personality.
In our sample there were no cases of deterioration of the psycho-emotional status during treatment. However, in cases with clinically significant depression and anxiety, patients are observed by gynecologists in conjunction with psychotherapists.
However, we associate the obtained data not only with the prescription of dienogest, but also with the complex management of the patients. It seems to us that the improvement of patient’s condition on the one hand is related to the diagnosis and confirmation/treatment of the disease during surgery. It is important for a girl and her parents to understand that after months (sometimes years, sometimes after several hospitalizations) and specialists (gastroenterologists, neurologists, surgeons, gynecologists) she receives final diagnosis and a practitioner who are ready to deal with the girl in a targeted manner. One this fact of the final diagnosis means that the girl has a chance to improve her condition.
On the other hand, the patient receives confirmation that the disease is not uncurable, that removes multiple fears (reproductive prognosis and others, that a teenager can come up without knowing the objective facts and having access to the Internet). It really seems to us that it is important to counsel such girls and their parents (who are often even more frightened than the girl) not only from a therapeutic, but also from a psychotherapeutic position, and that such guidance has an extremely positive effect.
Point 7. Line 413 /414: I do not agree. Ultrasound is another gold standard, especially in deep endometriosis. Please see recent ESHRE guideline, and recent publications on TVS and #Enzian.
Response 7. Thank you for this significant argument. Indeed, deep endometriosis with nodules more than 5 mm in diameter and endometriomas are well detected with imaging technics. But the big problem remains with superficial endometriosis, often seen in adolescents, and causing pronounced pain.
To date, we are starting to analyze ultrasound data with transrectal approach on initial stages of peritoneal endometriosis. Concerning transabdominal ultrasound we fail to detect lesions we usually see at laparoscopy (less than 3-5 mm in diameter) nor in the utero-sacral ligaments nor in the peritoneum of the douglas space. And transvaginal approach is inappropriate to most of adolescents.
Even according to MRI we can suspect initial stages only approximately in 80% of cases in adolescents, because the lesions are formally below the resolution of MRI. Still, we do our best to provide a tentative search for the presence of small T1-weighted hyperintense foci, to pull out grounds for the diagnosis of initial stages.
We tried to classify peritoneal endometriosis according to #Enzian scoring system, in most cases we have P1-P2 and B1-B2. We have no cases with involvement of ureter, rectum, diaphragm etc. In two cases there were endometriomas and in three adhesions in the tubo-ovarian unit.
We specified:
Line 219: In the studied sample the majority of adolescents were characterized with the I stage of endometriosis according to revised American Society for Reproductive Medicine (rASRM) score (43.8%, 14/32, the rASRM score 3.3±1.2 OR P1-2, O 0/0, T0/0, B1/0 C0 according to #Enzian(s) score). The rest of patients - with the II stage (34.4%, 11/32 the rASRM score 10.8±2.9 or P2, O 0/0, T0/0, B1/2, C0 according to #Enzian(s)) and the III stage of the disease (21.9%, 7/32, the rASRM score 20.4±7.6 or P2-3, B1/2 according to #Enzian(s) score). The III rASRM stage of the disease in two cases was observed on the background of adenomyomas (O2/0) and in three cases the adhesions on the tubo-ovarian unit (T1/0) were detected.
Point 8. Please shorten the conclusion. Maybe the results on dienogest treatment are the main aim of this work.
Response 8. Thank you very much for this fair comment. We shortened the conclusion, though we remained the factor analysis data of affective symptoms prediction, cause it seems to us rather significant in patients management and prognosis.
We thank You very much for your suggestions, critics and time to improve our paper!
Please see the attachment below!
